# Adding Value to Reclaimed Water from Wastewater Treatment Plants: The Environmental Feasibility of a Minimal Liquid Discharge System for the Case Study of Larnaca

Maria Avramidi *, Christina Spyropoulou, Constantinos Loizou, Maria Kyriazi, Jelica Novakovic, Konstantinos Moustakas 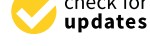, Dimitris Malamis and Maria Loizidou

Unit of Environmental Science and Technology, School of Chemical Engineering, National Technical University of Athens, Zographou Campus, 9 Heroon Polytechniou Street, 15780 Athens, Greece; chris.spyropoulou@gmail.com (C.S.); conloizou@hotmail.com (C.L.); kyriazimaria@mail.ntua.gr (M.K.); jelica@central.ntua.gr (J.N.); konmoust@central.ntua.gr (K.M.); dmalamis@chemeng.ntua.gr (D.M.); mloiz@chemeng.ntua.gr (M.L.)
* Correspondence: mariaavramidi@mail.ntua.gr

**Abstract:** The escalating water demand in Cyprus has necessitated the exploration of alternative water resources. The available water, which relies on rainfall and dam storage supplemented by methods such as desalination and aquifer enrichment, is inadequate to meet the current water demand. As a solution, Cyprus is utilizing reclaimed water for irrigation, in full compliance with both local and EU regulations. To address sustainable water management in Cyprus, a minimal liquid discharge (MLD) system is assessed for its environmental feasibility. A system incorporating reverse osmosis (RO), a multi-effect distillation (MED) evaporator, and a vacuum crystallizer (VC) is proposed for treating reclaimed water from the wastewater treatment plant (WWTP) in Larnaca. The proposed system aims to control the salinity (2500 mg/L) that limits the use of recovered water to the irrigation of non-sensitive types of crops, while recovering salt (sodium chloride). A life cycle assessment (LCA) was conducted, comparing the proposed MLD system with a reference system based on RO technology, where water is recovered, and brine is rejected back into the sea. The environmental feasibility was assessed via comparing 16 different environmental impact categories. Based on the analysis, the reference study provided a positive numeric value for most of the impact categories that were examined. Thus, it was concluded that the reference study has an overall negative impact on the environment, whereas the proposed MLD system demonstrated an overall positive impact, mainly due to low ecotoxicity.

**Keywords:** reclaimed water; wastewater treatment plant (WWTP); minimal liquid discharge (MLD); LCA; Larnaca



## 1. Introduction

In the coming years, there will be growing concern about the global decline in water supplies, which is attributed to several factors, including increased water demand, population growth, and industrial needs. The effects of climate change and rising temperatures further exacerbate the situation. Studies have shown an increase of about 3 trillion m³ in annual water consumption over a span of 64 years (from 1950 to 2014) [1]. Currently, around 20% of the global population lacks access to natural renewable water resources, while projections suggest that, by 2050, over half of the global population will experience severe water scarcity for at least part of the year [1–3]. In Europe, the total volume of supplied water reported in 2021 was around 45.9 billion m³/year, representing a 2.7% increase compared to the previous reporting year of 2017 [4].

According to the Food and Agriculture Organization of the United Nations (FAO), the expansion of land use areas is inextricably associated with the water scarcity issue [5].

In Europe, 40% of the total available water supply is needed for irrigation purposes [6]. According to the same study, around 4 billion m$^3$ and 91 million m$^3$ were used for irrigation in Greece and Cyprus, respectively [6]. The European Environment Agency (EEA) has proposed the Water Exploitation Index Plus (WEI+), which is a key indicator used to assess water scarcity conditions in Europe. Generally, it compares the use of freshwater resources with the available renewable water resources in a given area and time frame [7,8]. In terms of seasonal water scarcity, Cyprus, Malta, Greece, Portugal, Italy, and Spain encountered the most acute conditions among the EU-27 member states in 2019 [9]. In 2017, Cyprus recorded a WEI+ indicator of around 70%, a clear sign of significant water stress [10].

Cyprus has a population of around 1.2 million people, with the available exploitable water resource being approximately 0.54 km$^3$/year [11]. The area is characterized by high pressure, leading to restricted cumulative rainfall levels and extended winter drought periods [12]. The growing population, and Cyprus' economic shift toward a more resource-intensive consumption pattern, have been pivotal factors contributing to its water scarcity challenges. This challenge is exacerbated by the extensive overutilization of water in both occupational and residential activities throughout Cyprus [13,14]. Water demands fluctuate across different sectors. In 2020, the agricultural sector accounted for the largest share of water demand (59%) followed by household use (30%), with tourism (5%), farming (3%), and industry (3%) being the lowest water consumption sectors [15].

Over time, Cyprus has primarily relied on rainfall and stored water in dams. Other water sources from various sectors, such as desalination, recycled water, and aquifer enrichment, have made relatively minor contributions to the overall inflow [15]. For instance, the combined contribution of rainfall and water stored in dams accounted for 90% of the total water inflow in 2020 (Figure 1).

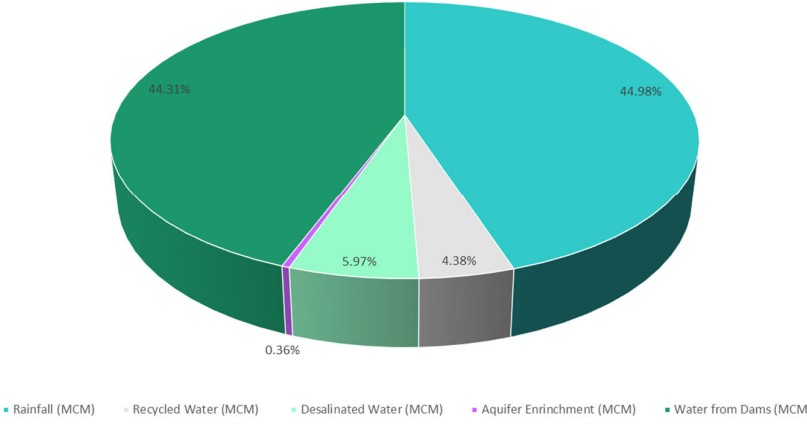

**Figure 1.** Total available water resources in Cyprus, in 2020 (% million cubic meters) [16–18].

Cyprus has 108 dams and artificial lakes, with a total capacity of around 332,000 Million cubic meters (MCM) [19]. Within this network, 18 major dams contribute substantially to this capacity, with a cumulative storage of 290,800 (MCM). However, during certain periods, such as 2012–2013 and 2015–2016, the inflow into these reservoirs barely exceeded 10%, underscoring these years as some of the most severe drought seasons in Cyprus [17]. In addition to its reliance on traditional water sources, Cyprus effectively employs recycled water to meet its remaining water needs. From 2010 to 2021, approximately 230 MCM of recycled water have been strategically utilized to enhance the region's water balance [16].

The utilization of recycled water from the WWTPs in Cyprus aligns seamlessly with the relevant guidelines and regulations. The Wastewater Directive (91/271/EEC), along with the EU Regulation 2020/741, promotes and facilitates water reuse in Cyprus [20,21]. The main objective of these regulations is to safeguard public health and aquifers from the detrimental effects caused by the disposal of inadequately treated urban sewage and its by-products, particularly sludge. According to (EU) 2020/741, "The reuse of treated urban wastewater for agricultural irrigation is a market-driven action" [21].

Cyprus has established regulations governing the recharge of its aquifers through Laws No. 106(I)/2002 and No. 127 of 2018 [22,23]. The utilization of recycled water for irrigation, on the other hand, is primarily overseen by the Code of Good Agricultural Practice (GAP) (No. 263/2007) [24]. Under this code, irrigation with recycled water can be employed for various crop types, including forest trees, fodder crops, fruit tree orchards, green areas, and most vegetables, apart from those typically consumed raw, such as leafy vegetables, bulbs, and condyles, such as lettuces, carrots, celery, and parsley [25]. According to the Water Development Department of Cyprus, some of the most extensively irrigated crops are corn, olives, Lolium, Sudex, various seasonal crops and, to a lesser extent, clover crops. Indicatively, 1.2 million $m^2$ of corn crops, and 378,000 $m^2$ of Lolium crops were irrigated with recycled water in 2022.

A good example of the reclamation and utilization of the recovered water from WWTP is Larnaca. Larnaca's WWTP stands as a significant facility, processing approximately 18,000 $m^3$/day, with a maximum capacity of 27,000 $m^3$/day. Annually, 3.5 million $m^3$ of tertiary treated water, along with 5000 $m^3$ sludge, is recovered at the Larnaca WWTP. A total of 70% of the available water is distributed to farmers for the cultivation of fodder plants, while 30% is available for the irrigation of gardens, fields, and other green spaces.

The available recycled water in Larnaca is distributed to 109 consumers and is used in a range of applications, including irrigation (77.6%), the maintenance of green areas by national and local authorities (12%), agricultural use in fields and hotel and home gardens (9.6%), and various other purposes (0.8%).

According to the data supplied by the Larnaca Sewerage and Drainage Board for the year 2022, the quantity of distributed recycled water fluctuated throughout the year, ranging from 79,060 $m^3$ in January to 390,000 $m^3$ in July. In February and March, the quantity increased nearly fourfold, reaching 319,935 $m^3$ and 340,810 $m^3$, respectively. From April to December, the distribution of recycled water dropped to almost 100,000 $m^3$, with the exceptions of July, August, and September, when the quantities remained at 300,000 $m^3$.

The only reported issue with the recovered water from Larnaca WWTP is the high salinity level. The average salinity of Larnaca's effluent is 2500 mg/L. Water with such high salinity levels is unsuitable for irrigating sensitive crops, while its long-term use can lead to salt accumulation in soil, along with direct and indirect adverse effects on the plants' health [26].

In this study, a minimal liquid discharge (MLD) was assessed, focusing on minimizing the water salinity to a level suitable for irrigating different types of crops and preserving soil quality. The system combines reverse osmosis (RO), a multi-effect distillation (MED) evaporator, and a vacuum crystallizer (VC), specifically designed to treat 1 $m^3$ of the reclaimed water from Larnaca's WWTP. This system not only aims to control the salinity, rendering the water suitable for a wide range of crops, but also focuses on recovering salt (sodium chloride), turning a waste product into a valuable resource. The environmental feasibility of the MLD system was assessed through LCA. Additionally, the MLD system was compared with a reference case, an RO system, which is the most used solution for water desalination.

## 2. Materials and Methods

### 2.1. Design and Development of the MLD System for Larnaca WWTP

A conventional zero liquid discharge (ZLD)/MLD system typically comprise three main stages: the pre-treatment, water recovery, and concentration stage [27]. The MLD system designed for this case study has been tailored to meet the requirements and challenges specific to Larnaca (Figure 2).

While various membrane technologies, such as FO (forward osmosis) [28,29], electrodialysis (ED) [30], membrane distillation (MD) [31], and pervaporation (PV) [32], are utilized in wastewater treatment and water purification, RO is the most frequently applied in the field. The optimization of operational conditions and energy-saving improvements throughout the years have led to an enormous increase in the establishment of

new plants [29]. RO can achieve energy efficiencies greater than 70% and low operational costs [33]. The RO units can treat different types of influents (brackish/seawater) [34], with a salinity varying from 0.20% *w/w* to 8.2% *w/w* [35].

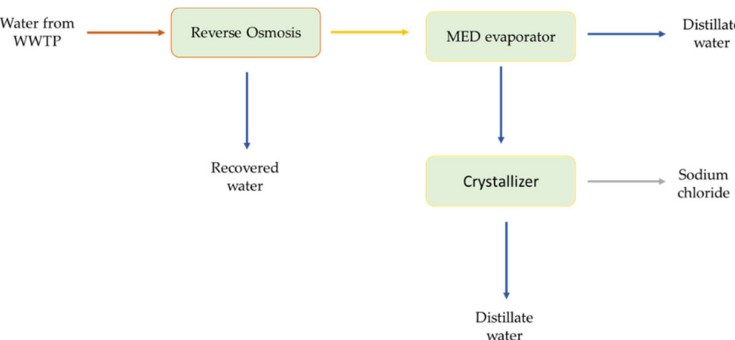

**Figure 2.** Process flow diagram of the proposed MLD system.

Different types of RO membranes exist on the market. The selection of the suitable one depends on the inflow volume, the required recovery, the rejection factor, and the quality of the product. For this case study, the membrane BW30-4040 was selected. This membrane is a spiral-wound element with a polyamide thin-film composite, and operates at low pressures, resulting in a low energy consumption while maintaining high salt rejection rates, above 83% [36].

The RO unit generates two streams, one of them being the produced water, and the other one being a high-salinity effluent, called brine. The generated brine has approximately twice the salinity of the influent [37]. The main disposal option for the brine is 45% surface discharge, 25% sewer discharge, 17% deep well injection, 7% land application, 5% evaporation, and 1% recycling [38]. However, these methods are considered unsustainable, as they are harmful to the environment and pose threats and risks to marine organisms due to the creation of a high-salinity environment [39]. The valorization of brines has been a great challenge in recent years, but MLD systems have emerged as a solution through incorporating a brine concentration step and recovering salt [40].

Various evaporation units, such as multi-effect distillation (MED) [41], multi-stage flash (MSF) [42], and mechanical vapor compression (MVC) exist on the market. These units play a key role in the concentration of the produced brine. Thermal units entail high energy demand due to the change of phase but, with proper design, operation, and maintenance, this demand can be minimized [43].

Regarding the final step in an MLD system, a crystallization process enables the recovery of different types of salts. The super-saturation of the effluents is essential to crystallizing the concentrated brine and can be achieved in various ways, according to the crystallizer type. There are several types of commercial crystallizers, and the selection of the most suitable depends on the desired end product and type of inflow. Options include batch or continuous crystallizers [44], which can utilize cooling, evaporative, or vacuum-based principles [45].

For this study, a MED evaporator and a VC were selected as, in a full-scale implementation, the two systems can be integrated using waste heat, thus significantly decreasing the energy cost.

### 2.2. Life Cycle Assessment of the Proposed System

In this research, the environmental impact of the MLD system process was assessed using life cycle assessment (LCA), according to the ISO 14040:2006 (EN) guidelines [46]. LCA involves four main phases, namely goal and scope identification, inventory analysis, impact assessment, and interpretation [46]. Data from the mass and energy balances of the various processes were used as the input for the SimaPro software (version 9.1.1.7), and the environmental impact was assessed using data from the Ecoinvent 3 database.

### 2.2.1. Goal and Scope

A comparative analysis was conducted between the two systems, the proposed MLD and the reference RO system, to assess their respective environmental impacts. The main objective was to evaluate the feasibility of the MLD system integrated into the WWTP facilities.

### 2.2.2. System Boundaries

As illustrated in Figure 3, the system boundaries for the first scenario include the RO unit with chemical, electricity, and auxiliary (membrane replacement) inputs. The chemicals consumed are hydrochloric acid, for pH adjustment and cleaning purposes, and sodium hydroxide, for fouling protection. The effluent of this system is the brine rejected into the environment. The process boundaries for the proposed MLD system are illustrated in Figure 4, with the required chemical, electricity, and auxiliary material input, and the recovered water and salt output.

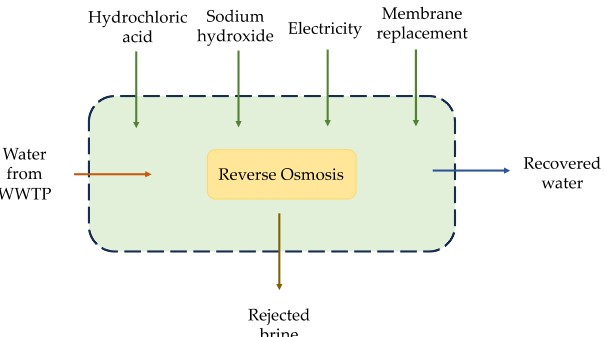

**Figure 3.** Reference case system boundaries.

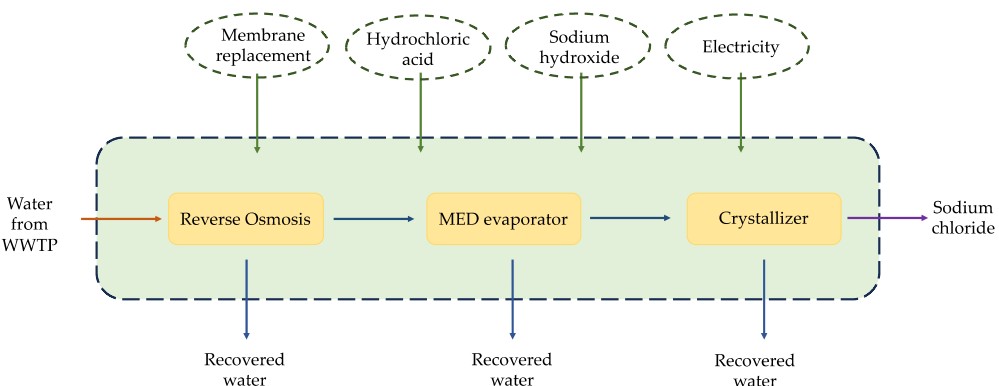

**Figure 4.** MLD case system boundaries.

All the data were modeled with the Ecoinvent database and the European Life Cycle Reference Database (ELCD). The Allocation at the Point of Substitution (APOS, S) system model was selected for the inputs [47]. The European (RER) level was selected for the geographical background of the added chemicals and the recovered water, while the Global (GLO) level was selected for the seawater reverse osmosis module and the rejected brine [48]. For the electricity demands, the high-voltage electricity data from Cyprus was used.

### 2.2.3. Inventory Analysis

For the two systems, the experimental data used in the LCA were derived from pilot systems at the Brine Excellence Center (BEC), located at the National Technical University of Athens, School of Chemical Engineering, Unit of Environmental Science & Technology. The pilot systems that are included in the BEC are an RO unit with a capacity of 0.5 m$^3$/h,

a two-effect MED evaporator with a capacity of 2 m$^3$/day, and a VC unit with a distillate water capacity production of 6.8 L/h. The tested RO unit comprises XLE-2540 elements that can also achieve high rejection factors, such as the BW30-4040. The two membranes are ideal for treating brackish water but the BW30 membrane can achieve higher rejections than the XLE membrane [49], which is why it was projected in this study. The BW30-4040 membrane has been tested in different scenarios for treating approximately 1 m$^3$ of brackish water with a varying number of membranes used, between 2 and 7 elements [50–52]. In the inventory analysis of the proposed systems, a 6-element RO system is considered, with a 7-year replacement rate for each membrane. The energy consumption and chemicals for pH adjustment and membrane cleaning were projected according to the results from the experiments conducted. These results were used for the evaporator and the crystallizer units, and the energy consumption of the systems was estimated as about 50 kWh/m$^3$ for the evaporator and 40 kWh/m$^3$ of recovered water for the crystallizer [27,53,54]. The projections of the MLD system are presented in Table 1.

**Table 1.** Flows of the MLD system.

| | MLD System | | | | | | |
|---|---|---|---|---|---|---|---|
| | **Reverse Osmosis Unit** | | | **MED Evaporator** | | **Crystallizer** | |
| | **WWTP Effluent** | **Recovered Water** | **Generated Brine** | **Recovered Water** | **Concentrate Stream** | **Recovered Water** | **Saturated in NaCl Stream** |
| K$^+$ (mg/L) | 48.77 | 1.31 | 238.61 | 2.49 | 1193 | 0.43 | 7158 |
| Na$^+$ (mg/L) | 555.7 | 14.81 | 2719 | 8.11 | 13,596 | 1.06 | 81,577 |
| Ca$^{+2}$ (mg/L) | 169 | 1.66 | 838.36 | 2.64 | 4191 | 0.69 | 25,150 |
| Mg$^{+2}$ (mg/L) | 69.18 | 0.7 | 343.1 | 0.25 | 1715 | 1.67 | 10,293 |
| Cl$^-$ (mg/L) | 784.8 | 20.27 | 3842 | 11 | 19,214 | 0.77 | 115,287 |
| SO$_4$$^{-2}$ (mg/L) | 634.9 | 6.08 | 3150 | 7.8 | 15,750 | 0.04 | 94,505 |
| HCO$_3$$^-$ (mg/L) | 249.2 | 7.334 | 1216 | 0.09 | 6083 | 0.96 | 36,499 |
| pH | 7.12 | 7.72 | 8.25 | 7.72 | 7.54 | 6.12 | 8.12 |

For the first process with the RO unit, the brine rejection into the environment is calculated with the net discharge of chemicals and ions to the sea [55], as brine is not included as a harmful waste in the SimaPro database.

The inventory analysis for both cases was calculated for 1 m$^3$ of inlet water from the WWTP of Larnaca, which is considered as the functional unit of the analysis (Table 2).

**Table 2.** Inventory analysis data.

| Inputs to LCA | Unit | 1 m$^3$ of Inlet |
|---|---|---|
| For the first process | | |
| HCl (30%) for the pH adjustment of RO | kg | 0.4025 |
| HCl (30%) for the membrane cleaning of RO | kg | 0.00335 |
| NaOH (50%) membrane cleaning of RO | kg | 0.00289 |
| Seawater reverse osmosis module | m$^2$ | 0.00057 |
| Recovered water from RO | kg | 800 |
| Sodium chloride, brine solution | kg | 2.46 |
| Electric energy of RO | kWh/m$^3$ | 0.8 |
| For the second process | | |
| HCl (30%) for the pH adjustment of RO | kg | 0.4025 |
| HCl (30%) for the membrane cleaning of RO | Kg | 0.00335 |
| NaOH (50%) membrane cleaning of RO | kg | 0.00289 |
| Seawater reverse osmosis module | m$^2$ | 0.00057 |
| Recovered water from RO | kg | 800 |
| Electric energy of RO | kWh/m$^3$ | 0.8 |

**Table 2.** *Cont.*

| Inputs to LCA | Unit | 1 m³ of Inlet |
|---|---|---|
| Electric energy of the MED evaporator | kWh/m³ | 50 |
| Distillate water from MED | kg | 160 |
| NaCl salt | kg | 2.52 |
| Distillate water from crystallizer | kg | 33.2 |
| Electric energy of the crystallizer unit | kWh/m³ | 40 |

## 3. Results and Discussion

### 3.1. The Life Cycle Impact Assessment (LCIA)

The outputs of the inventory data are analyzed with the LCD 2011 Midpoint+ method, which is used for the Product Environmental Footprint (PEF) and involves sixteen impact categories [56]. The results of the LCIA analysis are shown in Table 3. Positive output values indicate environmental burden or damage, while negative values indicate a possible positive environmental impact [57]. The result interpretation methodology aligns with other research on wastewater treatment and brine management systems, where two different case studies are compared with each other with the aim of evaluating their environmental impact [58,59]. Both the reference case study and the MLD system can contribute toward a solution to the water depletion problem due to their water recovery capability. The MLD system has 10 times higher $CO_2$ emissions compared to the reference case. The reference case has a more adverse effect in ozone depletion, ionizing radiation HH, and ionizing radiation E (interim), whereas the MLD system has a higher impact in the categories of particulate matter, photochemical ozone formation, acidification, terrestrial eutrophication, marine eutrophication, and land use. The key results of the hotspot impact categories are examined in the interpretation phase for their environmental effect.

**Table 3.** Overall impact assessment results.

| Impact Categories | | Reference Case | MLD System |
|---|---|---|---|
| Climate change | kg $CO_2$ eq | 8.111801582 | 76.93662 |
| Ozone depletion | kg CFC-11 eq | $8.74162 \times 10^{-6}$ | $2.12 \times 10^{-5}$ |
| Human toxicity, non-cancer effects | $CTU_h$ | $8.96984 \times 10^{-6}$ | $-1.3409 \times 10^{-5}$ |
| Human toxicity, cancer effects | $CTU_h$ | $1.13618 \times 10^{-6}$ | $-1.9511 \times 10^{-6}$ |
| Particulate matter | kg PM2.5 eq | 0.009104561 | 0.057195 |
| Ionizing radiation HH | kBq U235 eq | 1.144598178 | 0.468890 |
| Ionizing radiation E (interim) | $CTU_e$ | $3.64172 \times 10^{-6}$ | $2.28 \times 10^{-5}$ |
| Photochemical ozone formation | kg NMVOC eq | 0.029059083 | 0.332728 |
| Acidification | molc $H^+$ eq | 0.074622896 | 0.771287 |
| Terrestrial eutrophication | molc N eq | 0.107720009 | 1.13852496 |
| Freshwater eutrophication | kg P eq | 0.00641044 | −0.013632318 |
| Marine eutrophication | kg N eq | 0.009861996 | 0.09536647 |
| Freshwater ecotoxicity | $CTU_e$ | 900.8927124 | −1278.14923 |
| Land use | kg C deficit | 21.93650427 | 161.34076 |
| Water resource depletion | m³ water eq | −0.132457986 | −0.2211319 |
| Mineral, fossil, and renewable resource depletion | kg Sb eq | 0.00122319 | −0.001539 |

#### 3.1.1. Climate Change

The climate change indicator refers to the increase in the average global temperatures caused by greenhouse gas (GHG) emissions. As the MLD system incorporates more technologies, the total $CO_2$ emissions, as expected, were higher than those in the reference case. The main adverse environmental impact contributor to the MLD system is the high electricity consumption, while, for the reference case, it is the rejection of the brine solution. In terms of the electricity consumption impact category, the MLD accounts for 91.3480 kg $CO_2$ eq, a significantly high value compared to the 0.8047 kg $CO_2$ eq of the reference unit. The brine rejection factor for the reference case contributes about 90% of the

whole process's negative climate change effect (Table 4). The MLD system demonstrates a negative value attributed to the recovery of sodium chloride (Table 5). For the reference case, water recovery contributes to the negative value (positive impact) of the climate change impact category. Similar studies have highlighted the predominant impact of GHG emissions when developing a treatment system [60,61]. These studies demonstrate that, with the addition of technologies for the valorization of a discharge effluent, $CO_2$ emissions inevitably increase. For example, a ZLD system aiming to treat the saline effluent of a demineralized water plant (DWP) in the Netherlands reported that the addition of the ZLD system led to $CO_2$ emissions six times higher than those without its inclusion [61].

**Table 4.** The contribution of each input to the LCA in the examined impact categories for the reference case study.

| | Climate Change (kg $CO_2$ eq) | Human Toxicity—Cancer Effect ($CTU_h$) | Human Toxicity—Non-Cancer Effect ($CTU_h$) | Freshwater Ecotoxicity ($CTU_e$) | Mineral, Fossil, and Renewable Resource Depletion (kg Sb eq) | Freshwater Eutrophication (kg P eq) |
|---|---|---|---|---|---|---|
| Electricity, high voltage | 0.8047 | $2.44535 \times 10^{-8}$ | $2.66747 \times 10^{-8}$ | 1.4315 | $3.85123 \times 10^{-6}$ | $1.16798 \times 10^{-5}$ |
| Hydrochloric acid for pH adjustment | 0.2368 | $5.2337 \times 10^{-9}$ | $1.69088 \times 10^{-7}$ | 10.7306 | $1.50923 \times 10^{-5}$ | 0.0002 |
| Hydrochloric acid for cleaning | 0.0081 | $8.38407 \times 10^{-10}$ | $5.79731 \times 10^{-9}$ | 0.3679 | $5.17449 \times 10^{-7}$ | $6.52903 \times 10^{-6}$ |
| Seawater reverse osmosis module | 0.0499 | $7.21336 \times 10^{-11}$ | $3.42506 \times 10^{-10}$ | 0.0190 | $6.99749 \times 10^{-8}$ | $3.80574 \times 10^{-7}$ |
| Sodium chloride. Brine solution | 7.2769 | $1.23892 \times 10^{-6}$ | $8.95226 \times 10^{-6}$ | 893.8007 | 0.001211313 | 0.0064 |
| Sodium hydroxide | 0.0157 | $1.11977 \times 10^{-9}$ | $5.88437 \times 10^{-9}$ | 0.3538 | $5.34199 \times 10^{-7}$ | $8.36219 \times 10^{-6}$ |
| Recovered water from RO | −0.28041 | $-1.34463 \times 10^{-7}$ | $-1.90206 \times 10^{-7}$ | −5.8109 | $-8.18755 \times 10^{-6}$ | −0.0002 |

**Table 5.** The contribution of each input to the LCA in the examined impact categories for the MLD case.

| | Climate Change (kg $CO_2$ eq) | Human Toxicity—Cancer Effect ($CTU_h$) | Human Toxicity—Non-Cancer Effect ($CTU_h$) | Freshwater Ecotoxicity ($CTU_e$) | Mineral, Fossil, and Renewable Resource Depletion (kg Sb eq) | Freshwater Eutrophication (kg P eq) |
|---|---|---|---|---|---|---|
| Electricity, high voltage | 91.3480 | $5.94025 \times 10^{-7}$ | $3.02758 \times 10^{-6}$ | 162.4729 | 0.00044 | 0.0013 |
| Hydrochloric acid for pH adjustment and cleaning | 0.3907 | $4.11799 \times 10^{-8}$ | $2.62978 \times 10^{-7}$ | 15.9803 | $2.2037 \times 10^{-5}$ | 0.0004 |
| Seawater reverse osmosis module | 0.0499 | $7.21336 \times 10^{-11}$ | $3.42506 \times 10^{-10}$ | 0.0190 | $6.99749 \times 10^{-8}$ | $3.80574 \times 10^{-7}$ |
| Sodium hydroxide | 0.0101 | $9.98341 \times 10^{-10}$ | $5.13713 \times 10^{-9}$ | 0.3437 | $5.04259 \times 10^{-7}$ | $8.94393 \times 10^{-6}$ |
| Deionized water | 0.0431 | $-8.14836 \times 10^{-9}$ | $-3.16215 \times 10^{-8}$ | −1.8489 | $-2.39396 \times 10^{-6}$ | $-2.84081 \times 10^{-5}$ |
| Recovered water from RO | −0.2804 | $-1.34463 \times 10^{-7}$ | $-1.90206 \times 10^{-7}$ | −5.8109 | $-8.18756 \times 10^{-6}$ | −0.0002 |
| Sodium chloride | −14.6247 | $-2.44509 \times 10^{-6}$ | $-1.64832 \times 10^{-5}$ | −1449.3054 | −0.00199 | −0.0151 |

The higher energy consumption of the MLD system is attributed to the addition of technologies. Due to Cyprus' limited access to primary sources of energy, the Electricity Authority of Cyprus (EAC) relies entirely on imported fuels, primarily heavy fuel oil, to meet its electricity generation requirements. This reliance on fossil fuels for energy production contributes to air pollution through GHG emissions [62].

### 3.1.2. Human Toxicity

The human toxicity (non-cancer and cancer effects), measured in the comparative toxic unit for humans ($CTU_h$), reflects the potential impacts on human health resulting from the absorption of substances through the air, water, and soil. In total, the human toxicity effect of the MLD system has a negative value as opposed to the reference case, where the total value is positive. The negative value of human toxicity (non-cancer) is $-1.3409 \times 10^{-5}$ $CTU_h$, and for human toxicity (cancer) is $-1.9511 \times 10^{-6}$ $CTU_h$, both attributed to the sodium chloride recovery from the crystallizer unit. As for the reference case, the values are $+8.9698 \times 10^{-6}$ $CTU_h$ (non-cancer) and $+1.1362 \times 10^{-6}$ $CTU_h$ (cancer), respectively, with the only negative value deriving from the water recovery from the RO unit. Moreover, added chemicals and high electricity consumption have negative effects in the human toxicity category in both case studies.

The human ecotoxicity impact category, as highlighted in the study, [63] plays a major role in the total impact of the LCA analysis, when a recovery system process is not implemented. Another study, on the contrary, indicates that the process of reclaiming or treating water for reuse does not significantly reduce its potential to cause harm to human health compared to using the water in its untreated state [64].

### 3.1.3. Freshwater Eutrophication

The freshwater eutrophication factor serves as an indicator of the freshwater ecosystem enrichment with nutrient elements resulting from the release of nitrogen or phosphorus-containing compounds, and it is calculated from the LCA in kg phosphorus (P) eq. In total, the reference study has a positive value, indicating environmental burden. The main positive effect (99.8%) of the total value of 0.0064 kg P eq results from the brine discharge, with all the other factors contributing positively to the total value, except the water recovered from RO, with a value of $-5.811$ kg P eq. As for the MLD system, the overall impact is negative, indicating a positive effect on the environment. Most of the positive value is attributed to the use of NaOH for cleaning, accounting for a value of $8.3621 \times 10^{-6}$ kg P eq, whereas the overall negative value is associated with the recovered products, particularly the deionized water, making the most substantial contribution to the positive impact.

The effluent brine from seawater RO (SWRO) plants is characterized by a significant accumulation of salts and frequently includes different chemicals, like antiscalants based on phosphonates and coagulants based on ferric (or alum) sulfate, which are introduced during the desalination procedure [65]. Taking into consideration the possible phosphate and nitrogen load that could potentially enter the reclaimed water [66], the eutrophication possibility increases.

### 3.1.4. Freshwater Ecotoxicity

The freshwater ecotoxicity impact category relates to the potential toxic effects on an ecosystem, which can harm both individual species and the overall functioning of the ecosystem. In total, the MLD system has a positive environmental effect, with a negative value of $-1278.1$ CTUe, as opposed to the reference case, which has a negative effect, with a positive value of 900.90 CTUe. This significant difference, 377.2 CTUe, is due to the crystallizer and the sodium chloride recovery, as well as the fact that the reference case lacks any positive impacts, except the recovered water from the RO unit. The most significant negative impact of the MLD system is attributed to the high electricity consumption, with a value of +162.47 CTUe. The major contributor to the negative impact in the reference case is the brine discharge, with a value of +893.8 CTUe, similar to that in the study by Fayyaz [51], which is about six times higher than the most negative impact category of the MLD system. The added chemicals and the water recovery from both cases make a minor contribution to the negative impact of the systems.

### 3.1.5. Mineral, Fossil, and Renewable Resource Depletion

The impact category for mineral, fossil, and renewable resource depletion is relatively recent and has been introduced due to the decrease in climate change emissions. It is measured in kilogram of antimony (sb) equivalent (Kg Sb eq). In total, the MLD system has a positive effect on the environment, with a value $-0.00153$ kg Sb eq, as opposed to the reference case, which has a negative effect, with a positive value of $+0.001223$ kg Sb eq. The recovery of sodium chloride in the MLD system is the sole contributor to the total positive environmental effect of the system. The primary negative impact of the MLD system is associated with the high electricity consumption, totaling $+0.000437$ kg Sb eq. In contrast, the reference case shows that the major contributor to negative impacts is the brine discharge, accounting for $+0.00121$ kg Sb eq. The addition of chemicals has a negative impact on the environment, with tap water contributing positively to both cases.

In order to identify the contribution of each input and output parameter to the overall environmental impact, the absolute values were employed, which are presented in Figures 5 and 6 for both studied cases. For the reference case, the highest contribution to all the hotspot impact categories was the discharge of the brine solution. For the MLD system, sodium chloride recovery contributed to the overall positive impact of the proposed system, except for the climate change impact factor, where the high electricity consumption created a higher negative impact on the environment.

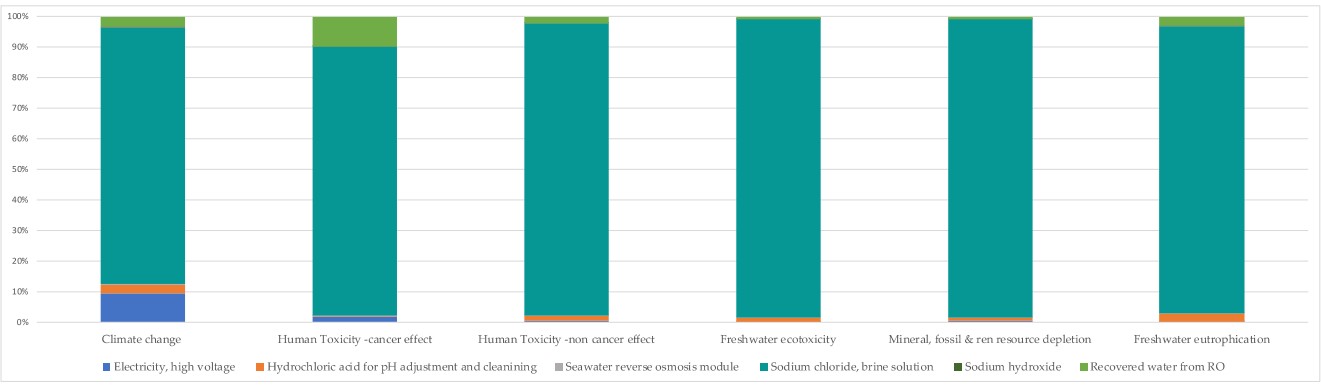

**Figure 5.** The contribution percentage of each input for the reference case on the hotspot impact factors.

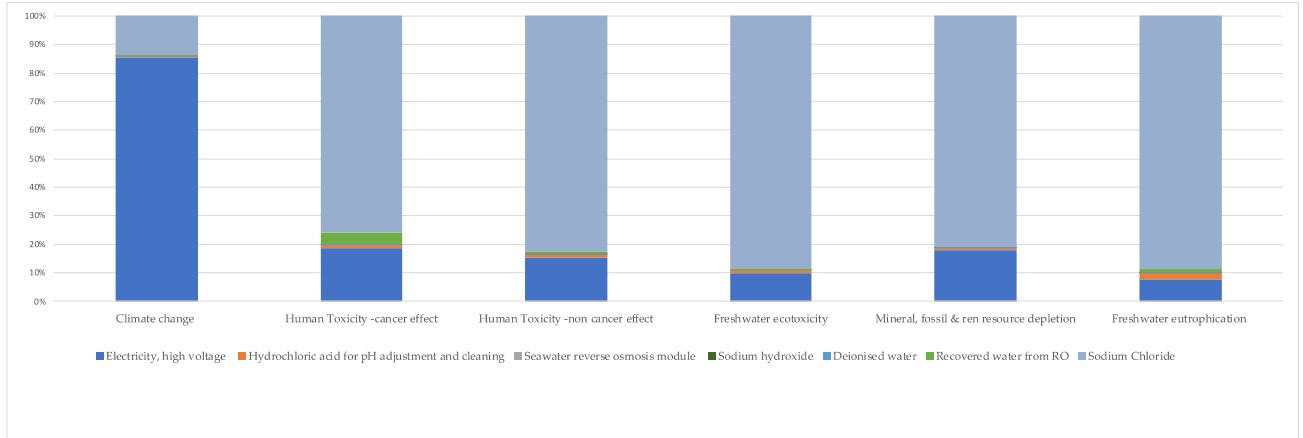

**Figure 6.** The contribution percentage of each input for the MLD system on the hotspot impact factors.

### 3.2. Sensitivity Analysis

The two most influenced parameters for the proposed MLD system were selected to identify the sensitivity of the conducted analysis. As presented in Figure 7, the required

electrical energy plays the most critical role in most environmental impact categories; thus, it was selected for the perturbation sensitivity analysis. Except for the energy requirement, sodium chloride recovery also affected the results of the LCA analysis, so both factors were analyzed.

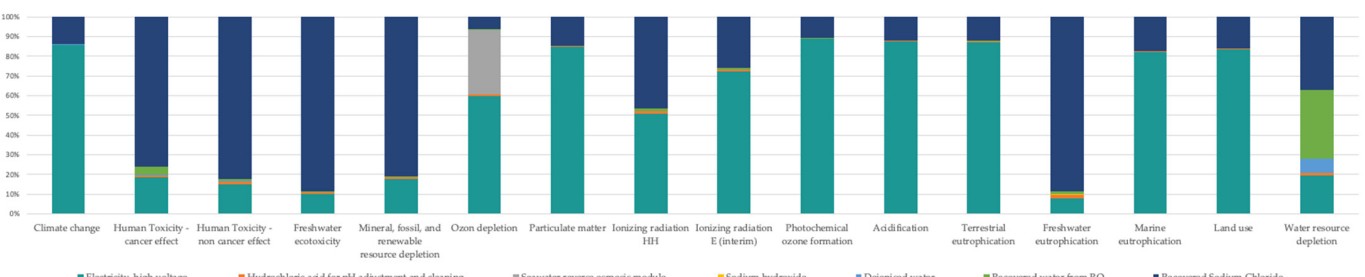

**Figure 7.** The contribution percentage of each input for the MLD system to all the impact factors.

The steps for conducting the perturbation analysis [57] are (1) defining the key parameters and initial values, (2) defining the environmental impact category, (3) conducting the sensitivity analysis where +10% and/or −10% variations of the defined key parameters can be implemented, (4) calculating the sensitivity ratio (SR) via the following formula, and finally (5) discussing the results and conclusions.

$$SR = \frac{\frac{\Delta result}{initial\ result}}{\frac{\Delta parameter}{initial\ parameter}} \tag{1}$$

where:

Δresult represents the difference between the results from the LCA to the +10% and −10% variation;

initial result represents the initial results of the value from the LCA before any variation;

Δparameter represents the difference between the input values between the +10% and −10% of the LCA of the selected parameter;

initial parameter represents the initial value of the parameter before any variations.

The sensitivity ratio can be used to assess how changes in different parameters affect the overall environmental impact of the process. The impact category examined as a reference was climate change, as all the impact categories have the same SR for the same parameter.

The total electrical energy consumption for the MLD system originates from the three integrated technologies, namely, the RO, the MED evaporator, and the VC. The sensitivity analysis was performed via changing the initial electricity consumption value by ±10% and examining the SR factor. Values of 99.8 kWh and 81.72 kWh were added to the SimaPro database to conduct the analysis. Accordingly, total recovered salt values were changed by ±10%, generating new outputs from the platform, and then they were analyzed.

The SR for the electrical energy was 0.88 and, for the NaCl recovery, around 0.98 absolute, indicating that the NaCl recovery has a higher impact on the overall environmental climate change factor.

Considering that electrical energy is a significant contributor to the environmental impact in all categories, a different type of sensitivity analysis was conducted to compare the impact of each system's energy consumption. For the conduction of this analysis, the energy consumption of each unit of the MLD system was calculated via changing the initial considered values by ±10%, and the SR factor was calculated. The SR factor for RO, MED, and VC was found to be 0.0054, 0.016, and 0.012, respectively. As expected, the MED imposed a higher burden on climate change, due to its higher energy demand.

## 4. Conclusions

An LCA was conducted between a reference case with brine discharge, and an MLD system where salt and water recovery are implemented, aiming at mitigating water resource depletion. The MLD system, while effectively addressing water scarcity, exhibits 10-times-higher $CO_2$ emissions than the reference case due to the increased energy consumption. However, the MLD system shows advantages in terms of its human toxicity level, freshwater ecotoxicity, and resource depletion. It offers potential improvements in human health and a positive environmental effect in the freshwater ecotoxicity category, mainly attributed to its crystallizer and sodium chloride recovery.

Electrical energy consumption stands out as the primary negative contributor to the MLD system's environmental impact, whereas in the reference case, brine discharge is the major source of negative impacts, being about six times higher than the most negative impact category of the MLD system.

The sensitivity analysis assesses how changes in various parameters affect the overall environmental impact. The SR for electrical energy consumption was calculated at 0.88, while, for NaCl recovery, it is approximately 0.98, indicating a higher impact from sodium chloride recovery. Given the significant role of electrical energy in the environmental impact across categories, a sensitivity analysis was conducted among the energy fluctuations of the MLD systems, revealing that the MED system, due to its higher energy demand, contributes more to the burden of climate change compared to RO and VC.

Considering these factors, the MLD system option appears to be more feasible than the conventional one, in terms of enhancing the quality of water produced by the wastewater treatment plant (WWTP). However, it is crucial to address the energy consumption issue and explore options to minimize it. Other solutions, such as incorporating renewable energy sources (RESs) or/and waste heat to power the system's operation, should also be examined.

**Author Contributions:** M.A.: Conceptualization, Methodology, Data Collection, and Data Analysis, Writing—Original Draft Preparation; C.S.: Data Analysis, Visualization, Writing; C.L.: Experimental set-up, Data analysis, Communication with Water Development Department (Cyprus), and Reviewing; M.K.: Writing—Review and Editing; J.N., K.M., D.M. and M.L.: Guidance, Supervision, Methodology, and Reviewing and Editing. All authors have read and agreed to the published version of the manuscript.

**Funding:** This research received no external funding.

**Institutional Review Board Statement:** Not applicable.

**Informed Consent Statement:** Not applicable.

**Data Availability Statement:** The authors affirm that the data supporting the findings of this study are accessible and provided within the article.

**Acknowledgments:** The authors would like to thank the Sewage Board of Larnaca and the Water Development Department of Cyprus for providing us with the analysis and distribution data for the reclaimed water in Cyprus.

**Conflicts of Interest:** The authors declare no conflict of interest.

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
