# Peer review of "Adding Value to Reclaimed Water from Wastewater Treatment Plants: The Environmental Feasibility of a Minimal Liquid Discharge System for the Case Study of Larnaca"

_sustainability, doi:10.3390/su151914305_

Round 1

Reviewer 1 Report

The manuscript compares the environmental impacts of two water recovery processes, RO and MLD, via LCA.  The outputs presented are valuable, especially in areas like Cyprus where significant water shortage is of concern. It truly deserves publication; however, there are some minor issues to be concerned.

-          Introduction is a bit long and contains some repetitions. Water availability/shortage issues in the study area could have been introduced more concisely.

-          Sec 2.1 would be moved and coupled to the Introduction section so that it will be a more holistic introduction.

-          Please check the title of Table 4 (inventory? or results?).

-          Please also check the numbers provided in Table 4 with those mentioned in the relevant discussion sections. Some of them seem not consistent with each other.

-          Please check the decimal separators used throughout the text. It seems some of them need to be replaced with “,”or “.”

-          Reference numbered 33 is not complete (year, volume, page??)

-          Sec 4.1 Interpretation phase. I think there is no need for this title. Could be removed.

-          Conclusion section could be shortened. It contains some statements that are more appropriate as of the introduction (e.g. first 3 paragraphs) 

Author Response

Dear Reviewer,

Reviewer 2 Report

I suggest to remove some figures eg. Fig. 1, 2 and 3 which are also described. 

The font in figures is to small -please correct particularly on figures 7 and 8. 

Author Response

Dear Reviewer,

Reviewer 3 Report

Title: Adding Value to reclaimed water from WWTPs: The environmental feasibility of an MLD system for the case study of Larnaca

Comments:

The manuscript sections are simply reports without in-depth analysis in the Introduction and results and discussion. Based on the current form, I would suggest a major revision for this manuscript. My suggestions are shown as follows:
1. The standard literature review process and analysis, for example, key word searching, paper screening, and publication analysis, need to be supplemented in the introduction section of the manuscript. There are many small paragraphs in the introduction section. However, there is a lack of interlinking between paragraphs.  Also present the novelty of the work and mention the objectives of the study clearly in the last paragraph.
2.
Combine results and discussion sections in a single section. The results and discussion section are very weak. The authors placed figures with a little bit of discussion on the results.  All the figures' quality must be improved. It is suggested to compare the results with literal values.

3. Conclusion is too big. The conclusion section should specify the outcomes of the study instead of discussing too much.

Moderate English improvement is required.

Author Response

Dear Reviewer,

Reviewer 4 Report

Dear Authors,

General Comments

1. Dont include any short forms in the title (Eg: WWTPs and MLD)

2. In some places, Grammer and language need to be improved.

Specific Comments

1.      Line 202, is mentioned as “Ρeverse” or Reverse

2.      In Inventory Analysis, which is Function and which is functional Unit?

3.      For Conducting LCA what are the inputs considered and how Environmental Impact is measured?

4.      Why were only two parameters considered for the sensitivity analysis?

5.      The present analysis applies to a global scenario or applies only to the selected location or region.

6.      The conclusion needs to be rewritten concerning the major findings and recommendations. The present conclusion part is like a discussion part.

Author Response

Dear Reviewer,

Reviewer 5 Report

Please see detailed review comments attached.

Needs improving further before the article can be published.

Author Response

Dear Reviewer,

Round 2

Reviewer 3 Report

Accept in current form

Reviewer 4 Report

Revision was satisfactory

Reviewer 5 Report

Thanks for making the changes. I had a read through the revised version and I'm still suggesting some more amendments - 

Abstract - the last sentence with result is too long, please can you split into smaller clearer sentences. "The environmental feasibility was assessed by comparing 16 environmental impact categories where, based on the analysis, was concluded that the reference study has an overall negative impact on the environment, based on the positive value of the majority of the impact categories that are examined, whereas the proposed MLD system demonstrated an overall positive impact to the environment mainly due to the negative value of the ecotoxicity environmental factor"

.

The language still needs further polishing to improve the clarity and readability of this paper. Needs more working to reach publishable standard.
